

**Role of black carbons mass size distribution in the direct aerosol radiative forcing**
Gang Zhao[1], Jiangchuan Tao[2], Ye Kuang[2], Chuanyang Shen[1], Yingli Yu[1], Chunsheng Zhao[1*]
[1]Department of Atmospheric and Oceanic Sciences, School of Physics, Peking University, Beijing,
China
[2]Institute for Environmental and Climate Research, Jinan University, Guangzhou 511443, China
*Correspondence to: Chunsheng Zhao (zcs@pku.edu.cn)*
**Abstract**

8       Large uncertainties exist when estimating radiative effects of ambient black carbon (BC) aerosol.

Previous studies about the BC aerosol radiative forcing mainly focus on the BC aerosols' mass
concentrations and mixing states, while the effects of BC mass size distribution (BCMSD) were not
well considered. In this paper, we developed a method by measuring the BCMSD by using a
differential mobility analyzer in tandem with an aethalometer. A comprehensive method of multiple
charging corrections was proposed and implemented in measuring the BCMSD. Good agreement
was obtained between the BC mass concentration integrated from this system and that measured in
bulk phase, demonstrating the reliability of our proposed method. Characteristics of the BCMSD and
corresponding radiative effects were studied based on field measurements conducted in the North
China Plain by using our own designed measurement system. Results showed that the BCMSD had
two modes and the mean peak diameters of the two modes were 150 nm and 503 nm respectively.
The BCMSD of coarser mode varied significantly under different pollution conditions with peak
diameter varying between 430 nm and 580 nm, which gave rise to significant variation in aerosol
buck optical properties. The aerosol direct aerosol radiative forcing was estimated to vary by 22.5%
for different measured BCMSDs, which shared the same magnitude to the variation associated with
assuming different aerosol mixing states (21.5%). Our study reveals that the BCMSD matters as well
as their mixing state in estimating the direct aerosol radiative forcing. Knowledge of the BCMSD
should be fully considered in climate models.
**1 Introduction**
Atmospheric black carbon (BC) is the second strongest absorbing components in atmosphere
(Bond et al., 2013) but the magnitudes of the warming effects are poorly quantified. When emitted to
the surrounding, BC particles transform the morphology from fractal to spherical and then grow as
fully compact particles with other components depositing on the BC aerosol (Peng et al., 2016). The





variation in the shapes of BC aerosols, together with the variation in the mixing states, can lead to
substantial change of aerosol optical properties (Liu et al., 2017;China et al., 2013;Wu et al.,
2016a;Wu et al., 2018). BC aerosols also have significant influence on the climate by interacting
with clouds (Koch and Del Genio, 2010;Roberts et al., 2008;Stevens and Feingold, 2009), ice and
snow (Bond et al., 2013). Recent study shows that the solar absorption of BC can suppress the
turbulence in the atmospheric boundary layer (Wilcox et al., 2016). It is found that BC emissions
may be responsible for the incensement of droughts and floods in China and India (Menon et al.,
2002). In addition, BC can pose a serve threat to human health through inhalation (Nichols et al.,
2013;Janssen et al., 2011).
Comprehensive studies have been carried out to evaluate the climate effect of BC based on the
measurement of BC mass concentrations ($m_{BC}$) (Koch et al., 2009;Ramanathan and Carmichael,
2008). The $m_{BC}$ near the ground have been well characterized (Ramachandran and Rajesh,
2007;Ran et al., 2016b;Reddington et al., 2013;Song et al., 2013), and the BC vertical distributions
are widely measured and evaluated as well (Ran et al., 2016a;Babu et al., 2011;Ferrero et al., 2011).
Despite these measurements, more insights into the BC microphysical properties can help to estimate
the influence of BC aerosols on visibility (Zhang et al., 2008), climate (Jacobson, 2001) and human
health (Lippmann and Albert, 1969). These microphysical properties include BC morphology (Zhang
et al., 2016), density (Zhang et al., 2016), complex refractive index (Bond et al., 2013), mixing states
(Moffet et al., 2016;Raatikainen et al., 2017), and particularly, the mass size distribution (BCMSD)
(Cheng et al., 2012;Cheng and Yang, 2016;Gong et al., 2016). Knowledge of BCMSD is not only
helpful to study the mixing state of BC aerosols (Raatikainen et al., 2017), but also essential to model
the role of BC in evaluating regional and global climate accurately (Huang and Yu, 2008b). BC
radiative effects is highly sensitive to the emitted BC particle size distribution (Matsui et al., 2018).
The health impacts of BC are significantly related to BCMSD (Turner et al., 2015). Furthermore, the
information of BCMSD can help to study the source, the evolution and the mixing state of ambient
BC aerosols (Yu et al., 2010). However, few studies have focused on the characteristics of the
BCMSD, and the BCMSD properties under different polluted conditions are not known yet.
Many methods have been proposed to measure the BCMSD. For instance, the BCMSD was
measured by sampling the aerosol in the size range from about 50 nm to several micrometers onto
quartz fiber filter substrates using a micro-orifice uniform deposit impactor (MOUDI) (Huang and



Yu, 2008b;Venkataraman and Friedlander, 1994;Guo, 2016). Cheng et al. (2014) developed a method
to measure the BCMSD by employing two aethalometers in parallel, with one to measure total $m_{BC}$
and the other to measure $m_{BC}$ below specific particle sizes using a size cut-off inlet. The Single
Particle Soot Photometer (SP2) is developed and widely used because it provides sing particle
information, hence the BCMSD and the mixing state of the atmospheric aerosols can be derived
directly (Schwarz et al., 2006;Gao et al., 2007;Huang et al., 2012;Singh et al., 2016). However, the
laser-induced incandescence method cannot provide reliable information about the particles beyond
the range of 70 nm and 400 nm (Moteki and Kondo, 2010), which results in the lack of the
knowledge of the BCMSD characteristics for these aerosols over 400 nm. The results from MOUDI
find that a great amount of BC locates at the diameter range larger than 370nm (Wang et al., 2015;Hu
et al., 2012). However, the measurements of MOUDI cannot give detailed information of the
BCMSD evolution due to the low temporal and diameter resolution (Xiaofeng Huang et al., 2006).
The characteristics of the BCMSD larger than 370 nm is not well studied due to the limitation of the
instrument.
Recently, Ning et al. (2013) and Stabile et al. (2012) proposed a new method to measure the
BCMSD by using differential mobility analyzer (DMA) in tandem with Aethalometer (AE). This
method has the potential of measuring the BCMSD from 20 nm to 584 nm with high time resolution.
We develop and validate the BCMSD measurement system based on the works of Ning et al. (2013).
The developed measurement system was employed in a field campaign in the North China Plain. The
characteristics of the measured BCMSD were studied based on the field measurement. Furthermore,
the effects of BCMSD variations on the aerosol optical properties and corresponding direct aerosol
radiative properties were evaluated. The aerosol optical properties were calculated by using the Mie
scattering theory. The direct aerosol radiative forcing (DARF) were estimated by using the Santa
Barbara DISORT (discrete ordinates radiative transfer) Atmospheric Radiative Transfer (SBDART)
model.
The structure of this paper are organized as follows. Section 2 gives the information about the
instrument setup and field measurement. Section 3 gives the detailed method used in this study,
which contains: 1, conducting multiple charging corrections when deriving the aerosol BCMSD and
2, evaluating the aerosol optical and radiative properties for different BCMSD. Results and
discussions are shown in section 4. The conclusion is drawn in the last part.



## 2 Instrument Setup

The measurement system setup was based on the works of Stabile et al. (2012) and Ning et al. (2013) as schematically shown in Fig.1. The ambient sample aerosol particles were firstly dried to below relative humidity of 30% through a Nafion drying tube before passing through to the DMA (Model 3081, TSI, USA). The DMA scanned aerosol particles with diameter ranges from 12.3 to 697 nm over a period of 285 seconds and started another scanning after a pause of 15 seconds, so one complete cycle took 5 minutes. The sheath and sample flow rates of the DMA were 3 lpm and 0.5 lpm, respectively. The quasi-monodisperse aerosols that passed through the DMA were further divided into two flows: with one lead to an aethalometer (AE51, Model 51, MicroAeth, USA) with a flow rate of 0.2 lpm to measure the $m_{BC}$ at 1 second time resolution; and the other one with flow rate of 0.3 lpm flow directed to a CPC (Model 3772, TSI, USA), which counted particle number concentrations at 0.1 second resolution. Clean air with a flow rate of 0.7 lpm was used to compensate for the CPC inlet flow, which had default flow rate of 1 lpm. Overall, the combination system of DMA, CPC and AE51 could provide one PNSD and BCMSD scan every 5 minutes.

At the same time, another aethalometer (AE33, Model 33, Magee, USA) was used to measure the $m_{BC}$ with a time resolution of 1 minute. The mass concentration of particles with diameter smaller than 2.5 μm (PM2.5) was concurrently measured with time resolution of 1 minute during the filed observations by the Tapered Element Oscillating Microbalance (TEOM) Dichotomous Ambient Particulate Monitor (1405-DF), which was an indicator of the pollution conditions.

From 21 March to 9 April in 2017, an intensive field measurement was conducted to characterize of the ambient aerosol BCMSD at the AERONET BEIJING_PKU station (N39°59′, E116°18′). This station was located on one roof of Peking University campus in the north west of Beijing, China. There were two main streets, Chengfu Road to the south and Zhongguancun Street to the west that surrounding the station. The aerosol sampled at this station were mainly composed of urban roadside aerosols (Zhao et al., 2018).

## 3 Methodologies

### 3.1 Retrieving the BCMSD

Four steps were involved to calculate the BCMSD using the raw data from the measurement system: 1), correcting the 'loading effect' of $m_{BC}$ measured by AE51; 2), matching the instrument time between the AE51 and CPC; 3), matching the measured $m_{BC}$ and diameter to get the raw





BCMSD that is not involved in multiple charging corrections; 4), conducting the multiple charging
corrections of the measured raw BCMSD.

### 3.1.1 Obtaining the raw BCMSD

The aethalometer (AE51 and AE33) is a well-developed and widely used instrument to measure
the $m_{BC}$ (Drinovec et al., 2015;Hansen et al., 1984). When absorbing aerosols accumulates on the
sample filter of the aethalometer continuously, the $m_{BC}$ can be determined by concurrently
measuring the light intensities $I$ after the fiber filter and the light intensities $I_0$ transmitted through
reference spot which is free of aerosol loading. The light attenuation (ATN) is defined as:
$$ATN = 100 \cdot \ln(\frac{I_0}{I}).$$    (1)
The mass of BC loaded on the filter is given by:
$$m_{BC,load} = \frac{A \cdot ATN}{100 \cdot \sigma_{BC}},$$    (2)
where A is the sample spot area on the filter and $\sigma_{BC}$ is the mass attenuation cross-section of BC.
The equivalent $m_{BC}$ can be calculated through the increment of $m_{BC,all}$:
$$m_{BC} = \frac{m_{BC,load}}{\Delta t} = \frac{A \cdot \Delta ATN}{100 \cdot \sigma_{BC} \cdot F \cdot \Delta t},$$    (3)
where F is the flow rate and ΔATN is the ATN variation during the time period of Δt.
Corrections of the measured $m_{BC}$ are necessary because the systematic bias exists due to the
prevailingly known 'loading effect' (Drinovec et al., 2015;Virkkula et al., 2015;Virkkula et al., 2007).
The AE33 can directly provide the corrected $m_{BC}$ values through measuring two light intensities of
two spots with different BC load efficiencies (Drinovec et al., 2015). For AE51, The correcting
method in Virkkula et al. (2007) was adopted:
$$m_{BC,corrected} = (1 + k \times ATN)m_{BC,uncorrected},$$    (4)
where k is the correction factor and a constant value of 0.004 is employed in this study to correct
the $m_{BC}$ from AE51. In the first part of the supplementary material, we showed that the loading
effects corrections of $m_{BC}$ from AE51 were essential and the value of $m_{BC}$ from AE33 could be
used as a reference for the measured BCMSD.
Time correction was needed because time gaps between voltages implied on the DMA (particle
size) and sample particles measured by different instruments were not the same. The time correction
procedures were conducted every day during the field measurement to ensure that the time deviations
of the CPC and the AE51 were constrained within 2 seconds.





Fig. S3 gave the time series diagram of scanned aerosol diameters by DMA, measured $m_{BC}$
from AE51, and the aerosol number concentrations counted by CPC. The aerosol PNSD (or BCMSD)
could be calculated by matching the DMA diameter and the measured aerosol number concentrations
(or measured $m_{BC}$) by simply using the single particle charge ratio for each electrical mobility
diameter. These measured PNSD and BCMSD did not consider the effect of multiple-charging
corrections and are labeled as raw aerosol PNSD and raw aerosol BCMSD.
**3.1.2 Multiple charging corrections of raw BCMSD**
In the work of Ning et al. (2013) study, lots of efforts were made to evaluate the performance of
the instrument. They considered the diffusion corrections and particle charging corrections. However,
the particle charging corrections were limited to single particle charge ratio as they mentioned that
they simplified the particle charge correction by applying the peak electrical mobility for the
calculation of representative particle size for each mobility bin and single particle charge ratio for
each primary mobility. They ignored the fact that the aerosol samples selected by the DMA were
quasi-monodisperse with different charges and different diameters.
We proposed a new algorithm for the multi-charge corrections of the BCMSD. Multi-charge
corrections to the measured size distribution were prevailing when the DMA was used to scan the
aerosol sizes. When the DMA and CPC are used together to measure the aerosol particle number size
distribution (PNSD), the multi-charging corrected aerosol PNSD can be significantly different from
the raw measured one (Bau et al., 2014;He and Dhaniyala, 2013;He et al., 2015). As shown in the
results part of this study, the multi-charge corrections of the BCMSD could cause differences in both
the magnitude and shape of the BCMSD. Therefore, it is necessary to perform multi-charge
corrections on the BCMSD. This study developed a new algorithm to correct the $m_{BC}$ from
measured $m_{BC}$ based on the work of Hagen and Alofs (2007) and Deng et al. (2011).
When the DMA is charged with a negative voltage, those aerosols with a small range of
electrical mobility ($Z_P$) can pass through the DMA:
$$Z_P = \frac{q_{sh}}{2\pi V L} \ln(\frac{r_1}{r_2}), \tag{5}$$
where $q_{sh}$ is the sheath air flow rate; V is the average voltage on the inner center rod; $r_1$ and $r_2$
are the outer and inner radius of annular space respectively. The $Z_P$ is related with $D_p$ by
elementary charge ($e$), number of elementary charges on the particle ($n$), and gas viscosity poise ($\mu$)



with:
$$Z_p = \frac{neC(D_p)}{3\pi\mu D_p},\qquad\qquad(6)$$
where $C(D_p)$ is Cunningham slip correction:
$$C = 1 + \frac{2\tau}{D_p}(1.142 + 0.558e^{-\frac{0.999D_p}{2\tau}}),\qquad(7)$$
where $\tau$ is the gas mean free path. From equation 7, aerosol particles can have the same $Z_P$ despite
that they have different $n$ and $D_p$. At the same time, there exists a relatively larger portion of
multiple charged particles for those particles with diameters between 100 nm and 400 nm when the
ambient aerosols pass through the X-ray (Tigges et al., 2015;Wiedensohler and Fissan, 1988).
Through the above discussion, the selected aerosols by DMA at a given electrical mobility can have
different charges which will correspond to different diameters.
When the scan diameter is set as $Dp_i$ for the singly charged particles and the respective voltage
of DMA is $V_i$ (i =1, 2, …, I), aerosol particles with electro-mobility of $Z_{p,i}$ (i=1, 2, …, I) can pass
through the DMA and the observed $m_{BC}$ by AE51 can be expressed as:
$$R_i = \int_0^\infty G(i,x)A(x)n(x)dx,\qquad\qquad(8)$$
where x is the scale parameter, with the definition of $x = \log(Dp_i)$, A(x) is the average BC mass
concentration of single particles for scale parameter x, and n(x) = dN/dlogDp is aerosol PNSD
that is the multiple charging corrected results from the measured aerosol PNSD. We define the kernel
function G (i, x), which is crucial to the algorithm, as:
$$G(i,x) = \sum_{v=1}^{\infty} \emptyset(x,v)\Omega(x,v,i),\qquad\qquad(9)$$
where $\emptyset(x,v)$ is the probability of particles that are charged with $v$ charges at the scale parameter
of $x$ (Wiedensohler, 1988). $\Omega(x,v,i)$ is the probability of particles that can pass through the DMA
with $v$ charges at the scale parameter $x$ (Knutson and Whitby, 1975). In this study, the maximum
value of $v$ is 10.
The multiple charging corrections can be expressed as computing the $A(x_i^*)$, in which $x_i^*$ is the
predetermined scale parameter from the DMA. To get the numerical integration results of equation 9,
the diameter interval that is 1/50 of the measured diameter is used. Thus, equation 9 can be written as
$$R_i = \int_0^\infty G(i,x)A(x)n(x)dx = \Delta x_i \sum_{j=1}^{50} \beta_j G(i,x_{i,j})A(x_{i,j})n(x_{i,j}),\quad(10)$$
where $\beta = \{\begin{matrix}0.5, & j=1,J \\ 1, & otherwith\end{matrix}$; $x_{i,j}$ is the $j^{th}$ (j=1 ,2 ,…, 50) parameter that locates at the parameter $x_i$ and





$x_{i+1}$ and $A(x_{i,j})$ (i=1, 2, …, I; j=1, 2, …, 50), the BC mass ratio at scale parameter $x_{i,j}$, is expressed
as the linear interpolation of the values at the measured diameters.
$$A(x_{i,j}) = A(x_i) + P_i(x_{i,j} - x_i),  \quad (11)$$
where $P_i$ is the slope of the linear interpolation result of
$$A(x_k^*) = C + P_i \cdot x_k^*.  \quad (12)$$
$x_k^*$ refers to these five diameters that are nearest to the predetermined scale parameter $x_i$. C is the
intercept of the linear interpolation result.
With $H_{i,j} = \beta_j \Delta x_i G(i, x_{i,j}) n(x_{i,j})$, equation 11 can be written as
$R_i = \sum_{j=1}^{J} H_{ij}[A(x_i) + P_i(x_{i,j} - x_i)] = \sum_{j=1}^{J} H_{ij}A(x_i) + \sum_{j=1}^{J} H_{ij} P_i x_{i,j} - \sum_{j=1}^{J} H_{ij} P_i x_i$
$= \sum_{k=1}^{I}(\sum_{j=1}^{J} H_{ij} \delta(i-k))A(x_k^*)) + \sum_{k=1}^{I}\left(\sum_{j=1}^{J} H_{ij} x_{i,j}\delta(i-k)\right)P_k - \sum_{k=1}^{I}\delta(i-k)))P_k x_k^*$
$= \sum_{k=1}^{I} Q_{ik}A(x_k^*) + \sum_{k=1}^{I} T_{ik}P_k - \sum_{k=1}^{I} Q_{ik}P_k x_k^*,  \quad (13)$
where $\delta(x) = \begin{cases} 0, x \neq 0 \\ 1, x = 0 \end{cases}$,
$$Q_{ik} = \sum_{j=1}^{J} H_{ij}\delta(i - k),  \quad (14)$$
and $T_{ik} = \sum_{j=1}^{J} H_{ij}x_{i,j}\delta(i - k).  \quad (15)$
By letting the
$$S_i = R_i - \sum_{k=1}^{I} T_{ik}P_k + \sum_{k=1}^{I} Q_{ik}P_k x_k^*.  \quad (16)$$
This equation is then expressed as
$$S_i = \sum_{k=1}^{I} Q_{ik}A(x_k^*),  \quad (17)$$
or
$$S = QA,  \quad (18)$$
where S and A are $I \times 1$ vectors and Q is an $I \times I$ matrix. This matrix can be solved by using the
non-negative least square method.
Finally, the A(x) can be determined and the corresponding BCMSD that is multiple charging
corrected can be calculated.
**3.1.3 Validation of the multiple charging corrections**
An example of the multiple charging corrections was shown in Fig. 2. The corrections of aerosol
PNSD were based on the work of Hagen and Alofs (2007). As shown in Fig. 2(a), the corrected





aerosol PNSD was significantly different from the original uncorrected one. There were about half of
the measured particles have multiple elementary charge in the size range between 100 and 200 nm.
The raw uncorrected aerosol PNSD had a peak value of 10920 cm$^{-3}$ at 98 nm while the corrected
aerosol PNSD reached its peak value of 8450 cm$^{-3}$ at 98 nm. The peak positions of the raw aerosol
particle mass size distribution (PMSD, dm/dlogDp) peaked at 371nm with a peak value of 56 μg/m$^3$
and the corrected aerosol PMSD had a peak value of 53 μg/m$^3$ at 445 nm. The peak position of the
aerosol PMSD shifted a lot before and after the multiple charging corrections. The similar case for
the BCMSD was shown in Fig. 2(b). The shape of BCMSD had changed substantially due to the
multiple charging corrections. The measured raw BCMSD had a peak diameter near 300nm and the
magnitude of BCMSD plateau reached 6000 ng/m$^3$ at 283nm, which was in accordance with the
results of Ning et al. (2013), where the multiple charging corrections were not involved. However,
the corrected BCMSD peaks near 400nm, with a peak value of about 5500 ng/m3 at 407nm.
According to the result, a small amount of BC remained in particles with diameter between 100nm
and 200nm. The measured BCMSD changed a lot when multiple charging corrections were
implemented, which highlighted the necessity of implementation of appropriate multiple charging
corrections

250        The $m_{BC}$ integrated from measured BCPMSD changed after multiple charging corrections. Fig.

S4 showed the comparison results of the $m_{BC}$ measured by AE33 and the $m_{BC}$ integrated from
AE51 measurements. The $m_{BC}$ integrated from uncorrected and corrected BCMSD versus $m_{BC}$
measured by AE33 were shown in Fig.S4(a) and Fig.S4(b), respectively. Before multiple charging
corrections, the $m_{BC}$ from uncorrected BCPMSD increased linearly with the $m_{BC}$ from AE33, with
R$^2$ equaling 0.87, but it was 2.37 times that of AE33 in average. As a comparison, overall magnitude
of $m_{BC}$ integrated from corrected BCPMSD agreed better with that from AE33 with a slope of 1.2.
With the discussion above, multiple charging corrections were essential for BCMSD measurements.
**3.2 Fitting the BCMSD by using two log-normal models**

259        Based on the measurement results, the BCMSD had two modes for most of the conditions. The

BCMSD are assumed to be of two log-normal distributions as:
$$m_{fit,Dp} = \sum_{i=1,2} \frac{m_i}{\sqrt{2\pi}\log(GSD_i)} \cdot \exp\left(-\frac{[\log(D_p)-\log(D_{m,i})]^2}{2\log^2(GSD_i)}\right), \quad (19)$$
Where $D_p$ is the diameter of the aerosols; $m_i$ is the mass of mode i (i=1,2); $GSD_i$ is the geometric





standard deviation at mode i (i=1,2), and $D_{m,i}$ is the geometric mean diameter of the mode i (i=1,2).
The $GSD_i$ and $D_{m,i}$ can be determined by using the least square method with the objective function
as :
$$J = \sum_{i=1,n}\left(m_{Dp_i} - m_{fit,Dp_i}(D_{m1}, GSD_1, D_{m2}, GSD_2)\right)^2, \qquad (20)$$
Where $m_{Dp_i}$ is the measured mass distribution at $Dp_i$, while $m_{fit,Dp_i}$ is the fit mass distribution at
$Dp_i$.
**3.3 Estimating aerosol optical properties with different BCMSD**
The Mie scattering model was used to study the influence of the BCMSD variation on the aerosol
optical properties. When running the Mie model, aerosol PNSD and BC were necessary. The aerosol
PNSD and $m_{BC}$ used here is the mean result of aerosol PNSD and $m_{BC}$ over the whole field
measurement respectively. The amount of BC particle adopted in this study is the mean value of the
$m_{BC}$ measured by AE33. In this study, The BCMSD was assumed to be log-normal distributed. $D_m$
of the BCMSD was set to vary from 100 nm to 600 nm. GSD of the BCMSD was set to be in the
range between 1.3 and 1.8. BC was treated as partially externally mixed and the remaining aerosols
was treated as core-shell mixed. The ratio of externally mixed $m_{BC}$ to core-shell $m_{BC}$ wea
determined by the method introduced in Ma et al. (2012) and a mean ratio of 0.51 was used. The
density and refractive index of BC were set as 1.5 g/cm3 and 1.8+0.54i (Kuang et al., 2015),
respectively. The complex refractive index of non-absorbing aerosols was 1.53+10-7i (Wex et al.,
2002) at the wavelength of 525 nm. More details of calculating the aerosol optical properties by
using the aerosol PNSD and BCMSD, can refer to Kuang et al. (2016). For each BCMSD, extinction
coefficient ($\sigma_{ext}$), the scattering coefficient ($\sigma_{sca}$), the single scattering albedo (SSA), and the
asymmetry factor (g) could be obtained from the output of Mie scattering model.
**3.4 Evaluating the DARF**
In this study, the SBDART model (Ricchiazzi et al., 1998) was employed to estimate the DARF.
In our study, the instantaneous DARF for cloud free conditions at the top of atmosphere was
calculated. Input of the model required the profiles of aerosol $\sigma_{ext}$, SSA, g. These values were
calculated by parameterized aerosol PNSD, BCMSD profiles. The corresponding DARF for different
BCMSD could be estimated. More details of estimating the DARF could refer to part 4 and 5 in the





supplementary material. The DARF was estimated for the measured mean aerosol PNSD and $m_{BC}$
under different BCMSD conditions to study the effects of BCMSD variations on the aerosol DARF.
**4 Results and Discussions**
**4.1 Measurement results of the BCMSD**
The time series of measured PM2.5, aerosol PNSD and BCMSD were shown in Fig. 3. During
the observation period, the PM2.5 varied from 0.06 to 220 μg/m$^3$, with a mean value of $71.5 \pm 52.56$
μg/m$^3$. Three periods of heavy PM2.5 loading were observed: (1) PM2.5 increased from around 100
μg/m$^3$ to 200 μg/m$^3$ and decreased slowly to 1 μg/m$^3$ in the period 21-26, March; (2) the PM2.5
accumulated slowly from 28 to 30, March and dissipated quickly from 30, March to 1, April; (3) the
rapid accumulation and dissipation of PM2.5 happened during 2 to 5, April. During the last five days,
PM2.5 fluctuated between 20 and 120 μg/m$^3$. For each pollution condition, both the aerosol total
number concentrations and the aerosol median diameter increased. The aerosol median diameter
varied between 31 nm and 169 nm with a mean value of $78 \pm 31$ nm.
As for the BCMSD, a distribution with two modes could be detected. The presence of the first
mode in the size range between 100 and 200 nm provided a verification of previous field
measurements that the BC concentrated in the particle diameter range from 100 to 200 nm. (Huang et
al., 2012;Ohata et al., 2011;Wu et al., 2016b). The peak diameter of second mode ranged from 300
nm to 600 nm, which agrees well with the measured BCMSD by MOUDI (KlausWilleke and Baron,
1996;Yu and Yu, 2009;Huang and Yu, 2008a). The main BC mass loading was in the coarser mode
for the sampled particles when comparing the BC mass concentrations at two modes.
The total $m_{BC}$ measured by AE33 ranged from 0.1 to 14 μg/m$^3$ with an average of $5.04 \pm 2.64$
μg/m$^3$. Good consistence was achieved between $m_{BC}$ measured by AE33 and $m_{BC}$ calculated from
measured BCMSD as shown in Fig. 4(c).
**4.2 Evolution of the BCMSD under different polluted conditions**
Log-normal distribution was used to fit each mode of the BCMSD by using the least square
method as introduced in section 3.2. For each mode, the geometric mean diameter ($D_m$) and the
geometric standard deviation (GSD) of the BCMSD were studied.
During the measurement period, both $D_m$ and GSD of the two modes had changed significantly
as shown in Fig 4. The $D_m$ of first and second mode varied from 128 to 162 nm and from 430 to



580 nm, respectively. The corresponding mean $D_m$ was 150 and 503 nm. The $D_m$ of the two
modes was found to be positive correlated in Fig. 4a. When the pollution was released from the
beginning to 27, March, the $D_m$ decreased from 590to 420 nm and from 155 to 130 nm for the
coarser mode and the smaller mode respectively. The BC containing aerosols tended to be aged and
grew larger when the air surrounding get polluted.

325        GSD for the coarser mode and the smaller mode showed very different properties as shown in

Fig. 4b. For the second mode, GSD varied from around 1.49 to 1.68 with a mean value of 1.57. The
GSD get decreased with the pollution condition, which indicated that BC containing aerosols tend to
accumulate to a small range of diameters during the aging processing. This phenomenon was
consistent with the fact that larger particles grew relative slower in diameter. For the first mode, GSD
ranged from 1.5 to around 1.85 with a mean value of 1.62. However, GSD of the smaller mode tend
to be larger when the surrounding air get cleaner, which might be related to the complex sources of
the BC emission. A small amount of fresh emitted BC particles can have substantial influence on the
mass size distribution of the smaller mode because the BC concentrations of the smaller mode were
small, especially under clean conditions. In general, the GSD of coarser mode was a good indicator
of the BC aging process and that of the smaller mode could partially reflect the complex sources of
the BC fine particles.

337        The relationship between the $D_m$ and the GSD for coarse mode was further analyzed by

analyzing the distribution of the $D_m$ and GSD. The GSD and $D_m$ had opposite trends as shown in
Fig 5. With the increment of the $D_m$ from 420 to 540 nm, the mean value of GSD decreased from
around 1.605 to 1.548 while the $m_{BC}$ increased with the $D_m$. The statistical relationship between
$D_m$ and GSD offered a reasonable representation of the BCMSD under different polluted conditions.
In the following work, mean values of the GSD at different $D_m$ were used to for further discussion.

343        Note that the GSD get slightly increased with the increment of $D_m$ when $D_m$ was larger than

520 nm. This might be caused by the limit diameter range of BCMSD measuring system which was
from 20 to 680 nm. The multiple charge corrections applied to the BCMSD could influence the
BCMSD when $D_m$ of the BCMSD was near the end of the scanned diameter and may lead to
significant uncertainties to the BCMSD. The measurement results indicated that cases of measured
$D_m$ of BCMSD larger than 520 nm were few, demonstrating that this multiple correction effect
influenced little on shape of measured BCMSD in most cases.





**4.3 Influence of BCMSD variation on the aerosol optical properties**
The aerosol optical parameters corresponding to different GSD and $D_m$ values were shown in
Fig. 6. In Fig. 6(a), the aerosol g varied from 0.617 to 0.649 (variation of 5.8%). Recent work by
Zhao et al, 2017 showed that the aerosol g value in the NCP may vary at a range of 10% due to the
change of aerosol PNSD. Aerosol g was more sensitive to $D_m$ when the geometric mean diameter of
the BCMSD was lower than 400 nm. However, when the $D_m$ was larger than 400 nm, the g become
sensitive to both the $D_m$ and the GSD of BCMSD. Overall, the g varied a little bit (0.617 to 0.624)
under the representative conditions during the measurement period. For the aerosol SSA, it was
sensitive to the $D_m$ over the whole range as shown in Fig. 6(b). SSA varied between 0.86 and 0.88
under the representative measurement conditions. The $\sigma_{sca}$ had large changes from 264 Mm$^{-1}$ to 313
Mm$^{-1}$. The $\sigma_{sca}$ was quite sensitive to variations in BCMSD when the $D_m$ was lower than 400 nm as
shown in Fig.6c, which varied substantially from 264 Mm$^{-1}$ to 313 Mm$^{-1}$. In addition, variations in
$\sigma_{sca}$ relied more on the variations in $D_m$ when $D_m$ was larger than 400 nm. Within the measurement
conditions of BCMSD, the $\sigma_{sca}$ varied from 265 Mm$^{-1}$ to 280 Mm$^{-1}$. The measured GSD under
different $D_m$ went along with the gradient direction of the $\sigma_{sca}$, which mean that the evolution of
BCMSD in the atmosphere influenced substantially on $\sigma_{sca}$. As for the $\sigma_{abs}$, it changed from 21.94
Mm$^{-1}$ to 44.12 Mm$^{-1}$ and the corresponding mass absorption cross section (MAC) was estimated to
be in the range of 4.75 to 9.56 m$^2$/g, suggesting that MAC of the BC aerosols should be carefully
studied under different BCMSD conditions.
**4.4 Influence of BCMSD on the direct aerosol radiative forcing**
The estimated DARF values for different GSD and $D_m$ conditions were estimated. When
estimating the DARF, the measured mean aerosol PNSD and mean BC mass concentration were
used . The results of estimated DARF were shown in Fig. 7(a). DARF at the surface varied from
-4.90 w/m$^2$ to -2.02 w/m$^2$ for different BCMSD. Within the measured BCMSD range, the DARF
varied from -2.04w/m$^2$ to -2.5w/m$^2$, which corresponding to 22.5% of variation. The heating rate
within the mixed layer was a powerful indicator of the BC particles' absorbing effects, which may
help evaluate the development of the boundary layer. The calculated mean heating rate within the
mixed layer changed from 3.25 K/day to 3.89 K/day for different $D_m$ and GSD, as shown in Fig.
7(b). The heating rate with the measured BCMSD range could change from 3.56 to 3.75 with a
variation of 5.23%.





Mixing states of BC play significant roles in calculations of aerosol optical properties and
estimations of DARF (Jacobson, 2001). As a comparison, we estimated the DARF under different
conditions of BC mixing state: (1) internally mixed, (2) externally mixed and (3) core-shell mixed.
Table 1 gave the estimated DARF and mean heating rate within the mixed layer under different
mixing state conditions. Results showed that the DARF under different BC mixing states conditions
may change by 21.5%, which shared the same magnitude with 22.5% variation of DARF caused by
BCMSD variations. In addition, the heating rate was estimated to vary by 6.05%. These results
highlighted that the BCMSD plays significant roles in variations of aerosol optical properties and
estimations of DARF as well as the air heating rate caused by the existence of BC particles. It was
recommended that a real time measured BCMSD be used when estimating the aerosol DARF, instead
of a constant one. The BCMSD was as important as that of the BC mixing states.
**5 Conclusions**
Knowledge of the BC microphysical properties especially the size-dependent information can
help reduce the uncertainties when estimating the aerosol radiative effects. BCMSD is an important
quantity in its own right, being directly and indirectly applicable to determination the sources, aging
processes and mixing states of BC aerosols. In this study, the characteristics of BCMSD were studied
from the field measurement results by using our own developed measurement algorithm.
The BCMSD measurement system was developed and validated based on the works of Ning et
al. (2013) by using differential mobility analyzer (DMA) in tandem with Aethalometer (AE). When
deriving the BCMSD, a comprehensive multiple charging correction algorithm was proposed and
implied. This algorithm was validated by closure study between the measured total $m_{BC}$ from AE33
and the $m_{BC}$ integrated from the measured BCPMSD using the datasets from field measurements.
Results showed that the multiple charging corrections could significantly change the shapes and
magnitudes of the raw measured BCPMSD. The accurate BCPMSD characteristics could be obtained
by our proposed method in this paper.
The developed measurement system was employed in a field campaign in the North China Plain
from 21 Match to 9 April in 2017. The BCMSD was found to have two quasi-lognormal modes with
peaks at around 150 nm and 500 nm, respectively. These two modes were consistent with the
previous measurement results by MOUDI (Wang et al., 2015;Hu et al., 2012). The amount of the BC
mass concentrations for the coarse mode peaks were about twice to that of the fine mode.





The characteristic of the BCMSD was studied by fitting the shape of BCMSD with a bi-normal
distribution. The relationships between the fitted $D_m$ and GSD were statistically studied. During the
aging processing, the opposite trends for the $D_m$ and GSD were found for coarser mode. This is the
first time that the coarser mode of the BCMSD were synthetically studied. The BCMSD of coarser
mode varied significantly under different pollution conditions with peak diameter changed between
430 and 580 nm. However, the relationship between the $D_m$ and GSD for smaller mode BC aerosols
were more complex due to the complex sources.
When the BCMSD were changed with the polluted condition, the corresponding aerosol optical
properties changes significantly. Sensitivity studies found that the aerosol g varies from 0.617 to
0.649 due to the variations in BCMSD. Aerosol g was more sensitive to $D_m$ when the geometric
mean diameter of the BCMSD is in the range of 300 nm and 370 nm. The SSA can changed from
0.86 to 0.93. The $\sigma_{sca}$ experienced significant change with the variation of BCMSD from 264 Mm$^{-1}$
to 313 Mm$^{-1}$ and the $\sigma_{abs}$ changed in the range between 21.94 Mm$^{-1}$ and 44.12 Mm$^{-1}$. The
corresponding BC MAC was calculated to be in the range between 4.75 and 9.56 m$^2$/g.
The variations in DARF were estimated due to the variations of the BCMSD by using the
SBDART model. Results showed that the DARF can varies by about 22.5% for different BCMSD
and the heating rate for different measured BCMSD conditions could change from 3.56 to 3.75,
corresponding to a variation of 5.23%. At the same time, the variations in DARF due to the
variations in the BC mixing state was estimated to be 21.5% and that of the heating rate is 6.05%.
Thus, the variations of the BCMSD may had significant influence on the aerosol radiative budget and
an accurate measurement of BCMSD was very necessary.

**Competing interests.** The authors declare that they have no conflict of interest.
**Data availability.** The data used in this study is available when requesting the authors.
**Author contributions.** GZ, CZ, JT and YK designed and conducted the experiments; CS, YY, CZ and GZ
discussed the results.
**Acknowledgments.** This work is supported by the National Key R&D Program of China (2016YFA0602001) and
the National Natural Science Foundation of China (41590872).

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





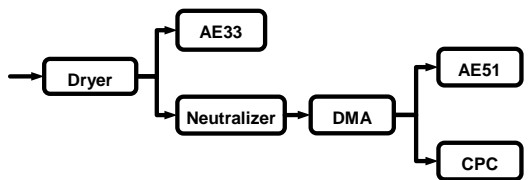


**Figure 1.** The schematic diagram of the instrument setup.





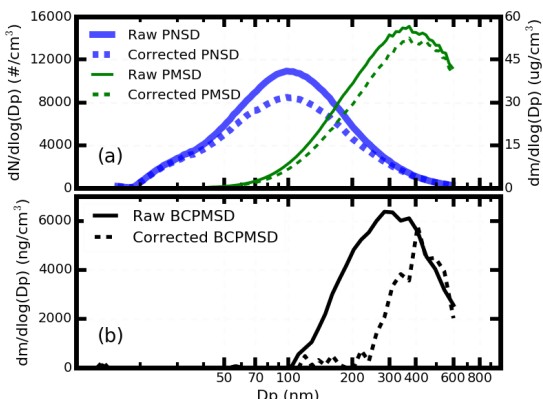


**Figure 2.** Case of multiple charging correction processing. (a) the multiple charging correction of the

aerosol PNSD and aerosol PMSD, (b) the multiple charging correction of the BCPMSD. The solid

line is the measured results without multiple charging corrections and the dotted line is the multiple

charging corrections results.







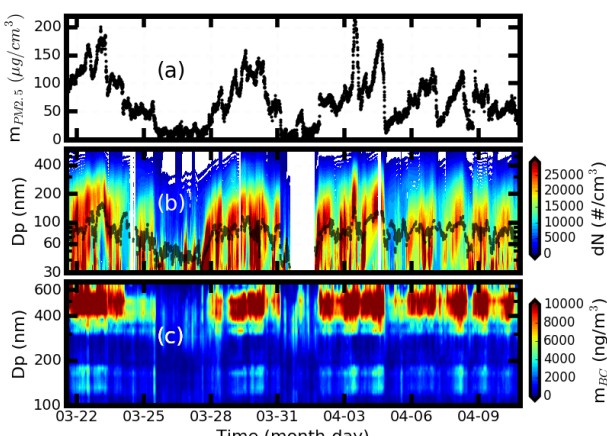


**Figure 3.** The measured time series of mass concentrations for (a) the PM2.5; (b) the aerosol

PNSD in filled color, the geometric median diameter in dotted line; and (c) the BCMSD.






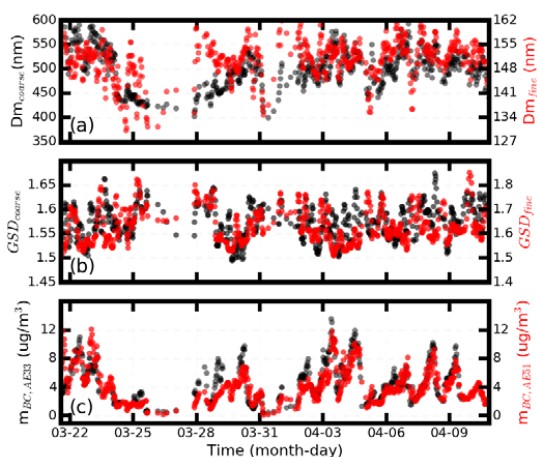


**Figure 4.** The (a) Dm and (b) GSD of the BCMSD at coarse mode (black) and fine mode (red); (c)
measured $m_{BC}$ by AE33 (black) and measured $m_{BC}$ from integrated $m_{BC}$ of the BCMSD from AE51.







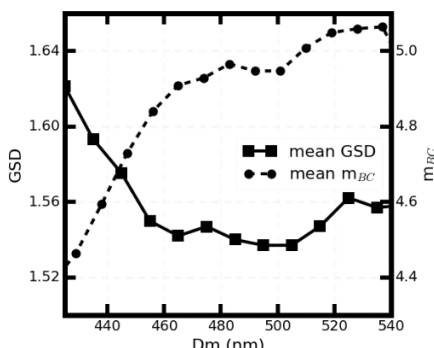


**Figure 5.** The relationship between the Dm and the GSD. The black dots show the real measured Dm
and GSD. The black line shows the mean results of the GSD for different Dm. The black line marked
with square shows the variation of mean $m_{BC}$ with the Dm.



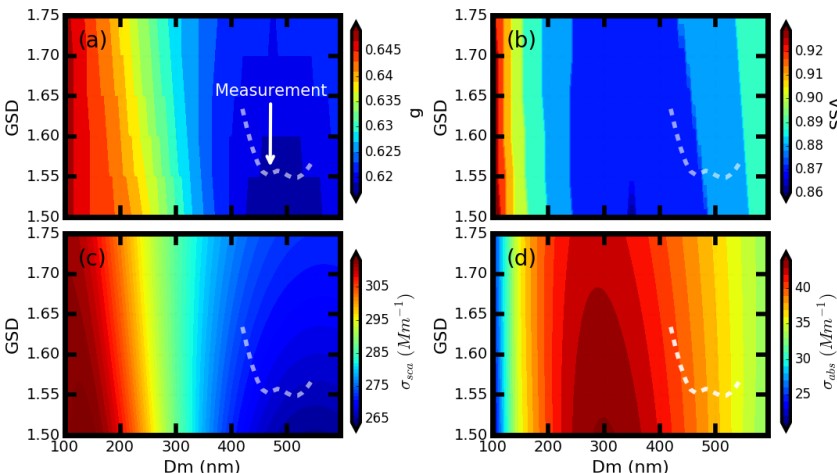

**Figure 6.** Variations of aerosol optics properties under different BCMSD conditions, which are
represented by different Dm and GSD values: (a) aerosol asymmetry factor, (b) single scatter albedo,
(c) scattering coefficient and (d) extinction coefficient . The grey dotted line in the figure shows the
evolution path of the BCMSD according to results of field measurements.





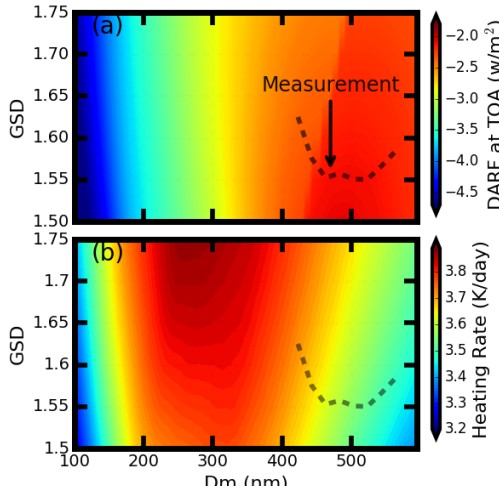


**Figure 7.** Variations of (a) DARF and (b) heating rate under different BCMSD conditions, which are

represented by different Dm and GSD values. The black dotted line in the figure shows the evolution

path of the BCMSD according to results of field measurements.






**Table 1.** Comparison of the DARF and heating rate values under different BC mixing states and
different BCMSD conditions.

| | | Mixing State | | | BCMSD | |
|---|---|---|---|---|---|---|
| | | Internal | External | Core-Shell | Minimum | Maximum |
| DARF | Value(w/m²) | -2.31 | -2.57 | -2.81 | -2.50 | -2.04 |
| | Variation | | 21.5% | | | 22.5% |
| Heat Rate | Value(K/day) | 3.67 | 3.47 | 3.68 | 3.56 | 3.75 |
| | Variation | | 6.05% | | | 5.23% |
