# Peer review of "Role of black carbons mass size distribution in the direct aerosol radiative forcing"

_Atmospheric Chemistry and Physics, 2019_

## Referee Comment (RC1) · Anonymous Referee #1 · 25 Jul 2019

This study uses combined two most common instruments to achieve the characterization of particle size for BC-containing particles, in addition addressing the importance of evaluating the radiative forcing impacts of BC by introducing the particle size. This study is well structured but needs to consider the following points before publication.

1. The observed two modes of the BCMSD in this study was the key conclusion, however the author should consider if the absorption measured by AE51 could be amplified due to the coating on BC particles at large size mode. As the aethalometer can only measure the absorption of the aerosol, it will bear large differences to convert the absorption of BC to BC mass at different size modes. The measured BCMSD will be also biased.

2. The radiative forcing impacts due to introducing the BC size information is rather

vague. As there is no direct measurement of vertical profile, how could SSA be so low in the upper level, which means there is a large fraction of BC, as shown in Fig. S6? It has not been demonstrated in the main text that how and why the BC size could influence the DARF results, due to the influence of asymmetry parameter? The DARF section needs a thorough revision I suggest.

Other comments

1. p. 1, line 23: the author should consider the BC mixing state on influencing the absorption at large mode.

2. p. 2, line 57: "and the BCMSD properties under different polluted conditions are not known yet", it was not correct since BCMSD have been measured using SP2 for years.

3. p. 2, line 58: BCMSD is not correct, the author should mention that the diameter measured by MOUDI, SP2 and DMA is different. The diameter for the MOUDI was aerodynamic diameter, and the DMA was the mobility diameter. BC diameter of the SP2 was the diameter of BC core. The value could change due to different density and shape factor. I strongly recommend the authors to rewrite this paragraph.

4. p. 3, line 72: there are some mistakes with the reference "(Xiaofeng Huang et al., 2006)".

5. p. 4, line 111: what does the "the ambient aerosol BCMSD" mean?

6. p. 5, Sec 3.1.1: The authors just mention how to correct the "loading effect" of the aethalometer, but not mentioning the multiple scattering correction of the filter which might overestimate the mBC value.

7. p. 5, line 145: If the measurement of AE33 haven't been corrected for the multiple scattering effect, it should not be treated as a reference.

---

## Referee Comment (RC2) · Anonymous Referee #2 · 29 Jul 2019

This paper reports a method combing a DMA with an aethalometer to obtain the BC mass size distribution (BCMSD). Two modes of the BCMSD are observed in their ambient measurement in the North China Plain with the new method. Also, they found that the BCMSD and their mixing state are equally important in estimating the aerosol direct radiative forcing, and suggested that the BCMSD should be fully considered in climate models. The method is a novel design and useful for understanding the relationship of BCMSD and their optical properties. The paper is interesting and well-organized. The conclusion is sound and may have profound impacts on the estimation of BC radiative forcing. I will recommend this manuscript for publication in ACP as long as the following comments are properly addressed.

Specific comments:

[Figure]

1. Line 245 and fig. 2. It seems that the shift of particle peak diameter for BCPMSD is much more significant than PMSD. Is there any explanation? The correction procedure should be the same for both. 2. Line 306 and fig. 3. It looks strange that there is nearly no BC in the size range of 200nm-300nm. Any explanation? What does the PMSD look like? To me, it looks like the consequence of overestimation of multiple charging of BC. If the data is confirmed to be correct, proper explanation is necessary. Besides, the authors cited several SP2 measurement to support BC peak diameter range from 100nm to 200 nm. However, if I understand correctly, SP2 measures the diameter of BC core, while this study measures the size of entire particles. More references of studies with other methods may be needed. 3. Line 48, I would add another BC microphysical property –hygroscopicity—in this sentence, which plays an important role in BC direct and indirect radiative forcing. (Zhang et al., 2008;Peng et al., 2017) 4. Line 275: Please add full name of GSD. 5. Line 277, "wea" should be "was" 6. Line 279, should the BC density differ with different mixing state assumption? Externally mixed BC normally exhibits lower density. The authors can considered this as future improvement direction. 7. Line 329, "larger particles grew relative slower in diameter." why? I would say larger particles grew slower in diameter in a log-normal scale. If this is what the authors meant, please verify. 8. Fig. 4 should be improved for better understanding by readership 9. Fig. 6, is the figures made with the assumption of the same particle number concentration? If so, please verify. 10. Table 1. The mixing state here is assumed to be internally mixed, externally mixed, and core-shell mixed. But what is the mixing state assumption for the estimation with BCMSD?

Peng, J. F., Hu, M., Guo, S., Du, Z. F., Shang, D. J., Zheng, J., Zheng, J., Zeng, L. M., Shao, M., Wu, Y. S., Collins, D., and Zhang, R. Y.: Ageing and hygroscopicity variation of black carbon particles in Beijing measured by a quasi-atmospheric aerosol evolution study (QUALITY) chamber, Atmospheric Chemistry and Physics, 17, 10333-10348, 2017. Zhang, R. Y., Khalizov, A. F., Pagels, J., Zhang, D., Xue, H. X., and McMurry, P. H.: Variability in morphology, hygroscopicity, and optical properties of soot aerosols during atmospheric processing, P Natl Acad Sci USA, 105, 10291-10296,

DOI 10.1073/pnas.0804860105, 2008.

---

## Author Comment (AC1) · 17 Sep 2019

Response to reviewer#1

Thanks for the reviewer's helpful suggestions! The point-by-point responses are listed below.

*Comment: This study uses combined two most common instruments to achieve the characterization of particle size for BC-containing particles, in addition addressing the importance of evaluating the radiative forcing impacts of BC by introducing the particle size. This study is well structured but needs to consider the following points before publication.*

**Reply:** We thank the reviewer's comments.

*Comment: 1. The observed two modes of the BCMSD in this study was the key conclusion, however the author should consider if the absorption measured by AE51 could be amplified due to the coating on BC particles at large size mode. As the aethalometer can only measure the absorption of the aerosol, it will bear large differences to convert the absorption of BC to BC mass at different size modes. The measured BCMSD will be also biased.*

**Reply:** We agree with the reviewer's valuable suggestion. Some revisions of deriving the BCMSD were made in the manuscript. In the manuscript, we first derived the BC absorption coefficient size distribution (BCASD), instead of the BCMSD. Then the BCASD were used to calculate the BCMSD using size-dependent mass absorption cross-section (MAC) value of BC.

The size-resolved MAC was calculated using the Mie scattering model (Bohren and Huffman, 2007). Based on the Mie scattering theory, the MAC varies for different aerosol core diameter and different aerosol diameter. Results from SP2 measurement show that the size distribution of the BC core diameter peaked at around 120 nm in Beijing (Zhang et al., 2017). For each aerosol diameter in our study, the MAC value with core diameter of 120 nm was used to transform the BCASD into the BCMSD. MAC values with core diameter at 120±15 nm were calculated and shown in Fig. R1. It varied significantly between 3.6 and 9.2 $m^2$/g.

Fig. R2 gave one example of the measured BCMSD using a constant MAC and a size-resolved MAC respectively. From fig. R2, the BCMSD was larger when the diameter was lower than 269 nm and smaller when the diameter was larger than 269 nm respectively when compared with the BCMSD derived using a constant MAC.

Some revisions were made in the manuscript.

[Figure]

**Figure R1.** Calculated mass absorption coefficient of different aerosol diameter with assumption that the core diameter is 120±15 nm

[Figure]

**Figure R2.** The retrieved mean BCMSD using a constant MAC of 7.7 g/m2 (solid line) and a variable MAC (dashed line) respectively.

*Comment: 2. The radiative forcing impacts due to introducing the BC size information is rather vague. As there is no direct measurement of vertical profile, how could SSA be so low in the upper level, which means there is a large fraction of BC, as shown in Fig. S6? It has not been demonstrated in the main text that how and why the BC size could influence the DARF results, due to the influence of asymmetry parameter? The DARF section needs a thorough revision I suggest.*

**Reply:** We thank the anonymous reviewer's comments and suggestions.

We added the multiple scattering correction of the measured light absorption coefficient in the revised manuscript. After the multiple scattering correction, the SSA was much larger than the previous value.

The BC size distribution influence the DARF results significantly. When the BCMSD was different for the same total BC mass concentration, the aerosol optical properties varied correspondingly. As noted in fig. 6 in the manuscript, the asymmetry factor, the scattering coefficient and absorption coefficient changed significantly. Therefore, the corresponding DARF were different due to different aerosol optical properties.

We added some discussions in the text. More descriptions of estimating the DARF were input in the manuscript.

*Comment: Other comments 1. p. 1, line 23: the author should consider the BC mixing state on influencing the absorption at large mode.*

**Reply:** Thanks for the comment. We have revised the text correspondingly.

*Comment: 2. p. 2, line 57: "and the BCMSD properties under different polluted conditions are not known yet", it was not correct since BCMSD have been measured using SP2 for years.*

**Reply:** We agree with the comment and deleted this sentence.

*Comment: 3. p. 2, line 58: BCMSD is not correct, the author should mention that the diameter measured by MOUDI, SP2 and DMA is different. The diameter for the MOUDI was aerodynamic diameter, and the DMA was the mobility diameter. BC diameter of the SP2 was the diameter of BC core. The value could change due to different density and shape factor. I strongly recommend the authors to rewrite this paragraph.*

**Reply:** Thanks for the comments. We rewrote this paragraph and added some descriptions in the text.

*Comment: 4. p. 3, line 72: there are some mistakes with the reference "(Xiaofeng Huang et al., 2006)".*

**Reply:** Thanks for the comment. We have revised the reference.

*Comment: 5. p. 4, line 111: what does the "the ambient aerosol BCMSD" mean?*

**Reply:** Thanks for the comment. We rephrased this sentence into 'ambient dry aerosol BCMSD corresponding to aerosol mobility diameter'.

*Comment: 6. p. 5, Sec 3.1.1: The authors just mention how to correct the "loading effect" of the aethalometer, but not mentioning the multiple scattering correction of the filter which might overestimate the mBC value.*

**Reply:** Thanks for the comments. We have revised the text correspondingly. As for the multiple scattering corrections, Zhang et al. (2018) compared the measured $\sigma_{abs}$ measured by AE33 and by Multi-Angle Absorption Photometer (MAAP) at Tsinghua University, which was about 2 km away from our measurement site. They recommended a compensation factor of 2.6. We used the same factor for multiple scattering correction in our study.

*Comment: 7 p. 5, line 145: If the measurement of AE33 haven't been corrected for the multiple scattering effect, it should not be treated as a reference.*

**Reply:** Thanks for the comment. We performed the multiple scattering effect corrections of AE33 in our manuscript.

[revised manuscript text omitted]

---

## Author Comment (AC2) · 17 Sep 2019

Response to reviewer#2

Thanks for the reviewer's helpful suggestions! The comments are addressed point-by-point and responses are listed below.

*Comment: This paper reports a method combing a DMA with an aethalometer to obtain the BC mass size distribution (BCMSD). Two modes of the BCMSD are observed in their ambient measurement in the North China Plain with the new method. Also, they found that the BCMSD and their mixing state are equally important in estimating the aerosol direct radiative forcing, and suggested that the BCMSD should be fully considered in climate models. The method is a novel design and useful for understanding the relationship of BCMSD and their optical properties. The paper is interesting and well-organized. The conclusion is sound and may have profound impacts on the estimation of BC radiative forcing. I will recommend this manuscript for publication in ACP as long as the following comments are properly addressed.*

**Reply:** We thank the reviewer's comments.

*Comment: Specific comments: 1. Line 245 and fig. 2. It seems that the shift of particle peak diameter for BCPMSD is much more significant than PMSD. Is there any explanation? The correction procedure should be the same for both.*

**Reply:** We thank the reviewer's helpful comments. We checked the fig. 2 (now fig. 3 in the revised manuscript) and made some revisions in fig 2.

There are three methods to calculate the PMSD. The first one is the corrected PMSD that is transformed directly from the multiple-charging corrected PNSD with multiple-charging correction. If the aerosol PNSD is $n(\log D_p) = \frac{dN}{d\log D_p}$, the corresponding PMSD after multiple-charging correction is:

$$\frac{dm}{d\log D_p} = n_V(\log D_p) \times \rho = V_p \times n(\log D_p) \times \rho = \frac{\pi}{6}\rho D_p^3 n(\log D_p)$$ (1).

PMSD calculated in this method corresponds to the corrected PMSD in fig. R1.

The measured PNSD without multiple-charging correction can be calculated from the multiple charging corrected PNSD as:

$$\left(\frac{dn}{dlogD_p}\right)raw = \frac{\sum_{v=1}^{\infty} n(x_v)\emptyset(x_v,v)}{\emptyset(x_1,1)} \qquad (2),$$

Where $x_v = logD_{p,v}$ (noting that $x_1 = logD_p$); $\emptyset(x_v, v)$ is the probability of particles that are charged with $v$ charges at the scale parameter of $x_v$; $\Omega(x, v)$ is the probability of particles that can pass through the DMA with $v$ charges at the scale parameter $x$. The relationship between the x and $v$ should meet the demand that their electrical mobility keep the same, ie

$$Z_p(D_{p,v}, v) = \frac{veC(D_{p,v})}{3\pi\mu D_{p,v}} = Z_p(D_{p,1}, 1) \qquad (3),$$

where $C(D_p)$ is Cunningham slip correction:

$$C(D_{p,v}) = 1 + \frac{2\tau}{D_{p,v}}(1.142 + 0.558e^{-\frac{0.999D_{p,v}}{2\tau}}) \qquad (4),$$

where $\tau$ is the gas mean free path.

The PMSD that corresponding to the measured PNSD can be calculated as

$$\left(\frac{dm}{dlogD_p}\right)_{raw} = \left(\frac{dV}{dlogD_p}\right)_{raw} \times \rho = \rho V_p \left(\frac{dn}{dlogD_p}\right)_{raw}$$

$$= \frac{\pi}{6}\rho D_p^3 (\frac{dn}{dlogD_p})raw = \frac{\pi}{6}\rho D_p^3 \frac{\sum_{v=1}^{\infty} n(x_v)\emptyset(x_v,v)}{\emptyset(x_1,1)} \qquad (5).$$

The PMSD calculated using this method corresponds to the calculated PMSD in the fig. R1.

The third method for PMSD measured by DMA-CPC system is calculated as:

$$\left(\frac{dm}{dlogD_p}\right)measure = \frac{\sum_{v=1}^{\infty}\frac{dm}{dlogD_{p,v}}\emptyset(x_v,v)}{\emptyset(x_1,1)} = \frac{\sum_{v=1}^{\infty}\frac{\pi}{6}\rho D_{p,v}^3 n(x_v)\emptyset(x_v,v)}{\emptyset(x_1,1)} \qquad (6).$$

Equation 5 can be transformed into:

$$\left(\frac{dm}{dlogD_p}\right)_{raw} = \frac{\sum_{v=1}^{\infty}\frac{\pi}{6}\rho D_p^3 n(x_v)\emptyset(x_v,v)}{\emptyset(x_1,1)} = \frac{\sum_{v=1}^{\infty}\frac{\pi}{6}\rho D_{p,1}^3 n(x_v)\emptyset(x_v,v)}{\emptyset(x_1,1)} \qquad (7).$$

As $D_{p,v}$ ($v=2,3,4$ ...) denotes the particle diameter with charged $v$ that shares the same electrical diameter with Dp of single charged particle. Therefore, $D_{p,v}$ ($v=2,3,4$ ...) is larger than $D_{p,1}$, and the relationship of these PMSDs using different methods is

$$\left(\frac{dm}{dlogD_p}\right)measure > \left(\frac{dm}{dlogD_p}\right)_{raw} > \frac{dm}{dlogD_p} \qquad (8).$$

From fig. R1, the multiple-charging corrected PMSD is significantly different from the measured PMSD. Both the magnitude and peak location are changed. As shown in fig. R2 (fig. 3 in the manuscript), the variations of the BCMSD and PMSD show almost the same pattern before and after the multiple charging corrections.

We revised the text and figure 3 in the manuscript.

[Figure]

**Fig. R1.** Example of measured PMSD (in dotted read line), calculated PMSD from raw PNSD (in dotted green line) and corrected PMSD (in green line)

[Figure]

**Figure R2.** Case of multiple charging correction processing. (a) the multiple charging correction of the aerosol PNSD and aerosol PMSD, (b) the multiple charging correction of the size-resolved $\sigma_{abs}$. The solid line is the measured results without multiple charging corrections and the dotted line is the multiple charging corrections results.

*Comment: 2. Line 306 and fig. 3. It looks strange that there is nearly no BC in the size range of 200nm-300nm. Any explanation? What does the PMSD look like? To me, it looks like the consequence of overestimation of multiple charging of BC. If the data is confirmed to be correct, proper explanation is necessary. Besides, the authors cited several SP2 measurement to support BC peak diameter range from 100nm to 200 nm. However, if I understand correctly, SP2 measures the diameter of BC core, while this study measures the size of entire particles. More references of studies with other methods may be needed.*

**Reply:** We thank the reviewer's comments. As described in reply of comment 1 and fig.R2, the multiple-charging corrections of the BCMSD is acceptable because the consequence of multiple charging of BCMSD is almost the same as that of the PMSD.

Our measurements show that the BCMSD has two modes with the coarser mode ranging between 430 nm and 580 nm in mobility diameter. Many field measurements have revealed that most of the BC mass locates in the aerodynamic diameter range of 320 nm and 560 nm using the MOUDI (Hu et al., 2012; Huang and Yu, 2008). When the aerodynamic diameter is transformed into mobility diameter with assumption a aerosol effective density of 1.3, the measured BC aerodynamic diameter range corresponds to mobility diameter range of 280 nm and 491 nm. Therefore, the measured size range for coarser mode of BCMSD agrees well with the previous measurement.

The measured aerosol in the field site is representative of the urban aerosol. The BC particles emitted by vehicles contribute significantly to the total aerosol BC mass. These BC particles are rarely coated or thinly coated, and the BC core diameter peaks around 120 nm (Zhang et al., 2017). Therefore, the BCMSD of the smaller mode measured in our study correspond to these uncoated of thinly coated particles.

We have revised the manuscript correspondingly.

*Comment: 3. Line 48, I would add another BC microphysical property hygroscopicity in this sentence, which plays an important role in BC direct and indirect radiative forcing. (Zhang et al., 2008;Peng et al., 2017)*

**Reply:** We thank the reviewer's comment. The manuscript has been revised correspondingly.

*Comment: 4. Line 275: Please add full name of GSD.*

**Reply:** We thank the reviewer's comment. We have revised the manuscript.

*Comment: 5. Line 277, "wea" should be "was"*

**Reply:** Thanks for the comment. We have changed it.

*Comment: 6. Line 279, should the BC density differ with different mixing state assumption? Externally mixed BC normally exhibits lower density. The authors can considered this as future improvement direction.*

**Reply:** We thank the reviewer's helpful suggestions. The reviewer provided a good view of the possible direction that we should focus on.

*Comment: 7. Line 329, "larger particles grew relative slower in diameter." why? I would say larger particles grew slower in diameter in a log-normal scale. If this is what the authors meant, please verify.*

**Reply:** Thanks for the comment. For the aerosol particles smaller than 1 um, they grow by collision and coalescence. Coalescence is much more efficient than collision (Lamb and Verlinde, 2011). For coalescence, the mass growth ratio is in proportion to the square of the diameter ($R^2$). However, the growth of the volume is in proportion to the cubic of diameter ($R^3$). Therefore, the growth ratio of aerosol particles is in proportion to the $R^{-1}$, which means that the larger particles grow slower than these of smaller particles in diameter do. We agree with the reviewer's idea that the larger particles appear to grow slower in a lognormal scale. We added some descriptions in the manuscript to denote it.

*Comment:* *8. Fig. 4 should be improved for better understanding by readership*

**Reply:** Thanks for the comment. We merged fig. 3 and part of fig. 4 into one figure.

*Comment:9. Fig. 6, is the figures made with the assumption of the same particle number concentration? If so, please verify.*

**Reply:** Thanks for the comment. The aerosol optical properties were calculated using the measured mean aerosol PNSD and different BCMSD. It was described in section 3.3. We added some descriptions in caption of fig. 6.

*Comment:10. Table 1. The mixing state here is assumed to be internally mixed, externally mixed, and core-shell mixed. But what is the mixing state assumption for the estimation with BCMSD?*

**Reply:** We thank the reviewer's comment. When estimating the radiative effects of BCMSD, we using the same mixing states of BC as that of Ma et al. (2012) in the North China Plain (NCP). Ma et al. (2012) found that the BC mixing states in NCP is partially core-shell mixed and partially externally mixed. The number ratio of the core-shell mixed BC particle and externally mixed particle is 0.51 Ma et al. (2012). We have added some descriptions in the manuscript.

[revised manuscript text omitted]

**5 Estimate the DARF**

DARF is defined as the difference between radiative flux at the TOA under present aerosol conditions and aerosol-free conditions:

$$\text{DARF} = (f_a \downarrow - f_a \uparrow) - (f_m \downarrow - f_m \uparrow) , \tag{21}$$

Where $f_a \downarrow$ is the downward radiative irradiance and $f_a \uparrow$ is the outward radiative irradiance under given aerosol distributions; $(f_a \downarrow - f_a \uparrow)$ is the downward radiative irradiance flux with given aerosol distributions and $(f_m \downarrow - f_m \uparrow)$ is the radiative irradiance flux under aerosol free conditions.

Input data for the SBDART are listed below. Vertical profiles of the aerosol optical properties, which include the aerosol extinction coefficient ($\sigma_{ext}$), aerosol single scattering albedo (SSA) and g with a height resolution of 50 m, come from the parameterization of aerosol vertical distributions (as shown in fig. S4 and the next paragraph) and the results of the Mie model. Atmospheric gas and meteorological parameter profiles come from the mean results of the radiosonde observations at the

Meteorological Bureau of Beijing (39°48' N, 116°28' E), which include profiles for water vapor, pressure and temperature during the spring. Surface albedo values are obtained from the Moderate

Resolution Imaging Spectroradiometer (MODIS) V005 Climate Modeling Grid (CMG) Albedo

Product (MCD43C3) during March, 2017 of Beijing, where the field campaign is conducted. The remaining input data for the SBDART are set to their default values.

**5.1 Parameterization of the aerosol vertical distribution**

Liu et al. (2009) studied vertical profiles of aerosol total number concentration (Na) with aircraft measurements, and derived a parameterized vertical distribution. In this scheme, Na is constant in the mixed layer, with a transition layer where it linearly decreases and an exponential decrease of Na above the transition layer. The same parameterized scheme proposed by Liu et al. (2009) is adopted by this study as shown in fig. S4 (b). Both the study of Liu et al. (2009) and Ferrero et al. (2010) manifest that the dry aerosol PNSD in the mixed layer varies little. The shape of the dry aerosol PNSD is assumed constant with height, which means that aerosol PNSD at different heights divided by Na give the same normalized PNSD.

As for the BC vertical distribution, Ferrero et al. (2011) and Ran et al. (2016) demonstrate that BC

mass concentration in the mixed layer remains relatively constant and decreases sharply above the mixed layer. According to this, the parameterization scheme of BC vertical distribution is assumed to be the same as that of aerosol. The shape of the size-resolved BC mass concentration distribution is also assumed to be the same as that at the surface.

[Figure]

**Figure S5.** The mean RH, temperature, and aerosol number concentration profiles.

**5.2 Calculate the aerosol optical profiles under the given RH profile**

With the vertical distribution of aerosol PNSD and BCMSD, the aerosol optical properties at a given RH profile can be calculated by using the Mie scattering model and κ-Köhler theory (Petters and Kreidenweis, 2007).

The aerosol hygroscopic growth is taken into consideration when calculate the aerosol optical properties under the given RH. The κ-Köhler theory (Petters and Kreidenweis, 2007) is widely used to describe the hygroscopic growth of aerosol particles by using a single aerosol hygroscopic growth parameter (κ) and the κ-Köhler equation, which is shown as

$$\frac{RH}{100} = \frac{gf^3 - 1}{gf^3 - (1-\kappa)} \cdot \exp\left(\frac{4\sigma_{s/a} M_{water}}{R \cdot T \cdot D_d \cdot gf \cdot \rho_w}\right) \ , \tag{1}$$

where $D_d$ is the dry particle diameter; gf(RH) is the aerosol growth factor, which is defined as the ratio of the aerosol diameter at a given RH and the dry aerosol diameter ($D_{RH}/D_d$); T is the temperature; $\sigma_{s/a}$ is the surface tension of the solution; R is the universal gas constant and $\rho_w$ is the density of water. The aerosol hygroscopic growth parameter κ can be further used to investigate the influence of aerosol hygroscopic growth on aerosol optical properties (Tao et al., 2014;Kuang et al., 2015;Zhao et al., 2017) and aerosol liquids water contents (Bian et al., 2014).

The κ-Köhler theory and the Mie scattering model are combined to calculate aerosol extinction coefficient, aerosol single scattering albedo and aerosol asymmetry factor under different RH conditions. The measured mean κ, which is derived from the humidified nephelometer system (Kuang et al., 2017), is used to account for aerosol hygroscopic growth. For each RH value, the gf can be calculated based on equation (1). The corresponding ambient aerosol PNSD at a given RH can be determined. The refractive index ($\tilde{m}$), which accounts for water content in the particle, is derived as a volume mixture between the dry aerosol and water (Wex et al., 2002):

$$\tilde{m} = f_{V,dry}\,\tilde{m}_{aero,dry} + (1 - f_{V,dry})\,\tilde{m}_{water} \qquad (2)$$

where $f_{v,dry}$ is the ratio of the dry aerosol volume to the total aerosol volume under a given RH

condition; $\tilde{m}_{aero,dry}$ is the refractive index for dry ambient aerosols and  $\tilde{m}_{water}$, the refractive index of water, is $1.33+10^{-7}$i. Then, the corresponding aerosol optical properties under the given RH and

PNSD can also be calculated. Finally, the aerosol optical profiles can be calculated. Fig. S6 shows one of the calculated aerosol optical profiles.

[Figure]

**Figure S6.** The calculated profiles of the aerosol extinction coefficient, aerosol single scattering albedo and the aerosol asymmetry factor.

**6 Relationship between the GSD, Dm and m$_{BC}$**

[Figure]

**Figure S7.** The (a) Dm and (b) GSD of the BCMSD at coarse mode (black) and fine mode (red); (c)

measured $m_{BC}$ by AE33 (black) and measured $m_{BC}$ from integrated $m_{BC}$ of the BCMSD from AE51.

Bian, Y. X., Zhao, C. S., Ma, N., Chen, J., and Xu, W. Y.: A study of aerosol liquid water content based on hygroscopicity measurements at high relative humidity in the North China Plain, Atmospheric Chemistry and Physics, 14, 6417-6426, 10.5194/acp-14-6417-2014, 2014.

Ferrero, L., Perrone, M. G., Petraccone, S., Sangiorgi, G., Ferrini, B. S., Lo Porto, C., Lazzati, Z., Cocchi, D., Bruno, F., Greco, F., Riccio, A., and Bolzacchini, E.: Vertically-resolved particle size distribution within and above the mixing layer over the Milan metropolitan area, Atmospheric Chemistry and Physics, 10, 3915-3932, 2010.

Ferrero, L., Mocnik, G., Ferrini, B. S., Perrone, M. G., Sangiorgi, G., and Bolzacchini, E.: Vertical profiles of aerosol absorption coefficient from micro-Aethalometer data and Mie calculation over Milan, Science of the Total Environment, 409, 2824-2837, 2011.

Kuang, Y., Zhao, C. S., Tao, J. C., and Ma, N.: Diurnal variations of aerosol optical properties in the North China Plain and their influences on the estimates of direct aerosol radiative effect, Atmos. Chem. Phys., 15, 5761-5772, 10.5194/acp-15-5761-2015, 2015.

Kuang, Y., Zhao, C., Tao, J., Bian, Y., Ma, N., and Zhao, G.: A novel method for deriving the aerosol hygroscopicity parameter based only on measurements from a humidified nephelometer system, 
[revised manuscript text omitted]